# Revised Rainfall Threshold in the Indonesian Landslide Early Warning System

Ragil Andika Yuniawan [1,2], Ahmad Rifa'i [1,*], Fikri Faris [1], Andy Subiyantoro [3,4], Ratna Satyaningsih [3,5], Alidina Nurul Hidayah [2], Rokhmat Hidayat [2], Akhyar Mushthofa [2], Banata Wachid Ridwan [2], Eka Priangga [2], Agus Setyo Muntohar [6], Victor G. Jetten [3], Cees J. van Westen [3], Bastian V. den Bout [3] and Samuel J. Sutanto [7]

1    Department of Civil and Environmental Engineering, Faculty of Engineering, Universitas Gadjah Mada, Jalan Grafika No. 2, Yogyakarta 55281, Indonesia; ragilandika@mail.ugm.ac.id (R.A.Y.); fikri.faris@ugm.ac.id (F.F.)
2    Balai Teknik Sabo, Direktorat Bina Teknik, Ministry of Public Works and Housing, Jl. Sabo No.1, Maguwoharjo, Sleman, Yogyakarta 55282, Indonesia; alidina.hidayah@pu.go.id (A.N.H.); rokhmat.hidayat@pu.go.id (R.H.); akhyar.mushthofa@pu.go.id (A.M.); banata.wr@pu.go.id (B.W.R.); ekapriangga@gmail.com (E.P.)
3    Faculty of Geo-Information Science and Earth Observation (ITC), University of Twente, 7514 AE Enschede, The Netherlands; andysubi@pu.go.id (A.S.); ratnasat@gmail.com (R.S.); v.g.jetten@utwente.nl (V.G.J.); c.j.vanwesten@utwente.nl (C.J.v.W.); b.vandenbout@utwente.nl (B.V.d.B.)
4    Balai Besar Wilayah C3, Direktorat Jenderal Sumber Daya Air, Ministry of Public Works and Housing, Jl. Ustad. Uzair Yahya No.1. Serang, Banten 206111, Indonesia
5    Center for Research and Development, Indonesian Agency for Meteorology, Climatology and Geophysics, Jakarta 10720, Indonesia
6    Department of Civil Engineering, Universitas Muhammadiyah Yogyakarta (UMY), Yogyakarta 55183, Indonesia; muntohar@umy.ac.id
7    Water System and Global Change Group, Wageningen University and Research, Droevendaalsesteeg 3a, 6708 PB Wageningen, The Netherlands; samuel.sutanto@wur.nl
*    Correspondence: ahmad.rifai@ugm.ac.id

**Abstract:** Landslides are one of the most disastrous natural hazards that frequently occur in Indonesia. In 2017, Balai Sabo developed an Indonesia Landslide Early Warning System (ILEWS) by utilizing a single rainfall threshold for an entire nation, leading to inaccuracy in landslide predictions. The study aimed to improve the accuracy of the system by updating the rainfall threshold. We analyzed 420 landslide events in Java with the 1-day and 3-day effective antecedent rainfall for each landslide event. Rainfall data were obtained from the Global Precipitation Measurement (GPM), which is also used in the ILEWS. We propose four methods to derive the thresholds: the first is the existing threshold applied in the Balai Sabo ILEWS, the second and third use the average and minimum values of rainfall that trigger landslides, respectively, and the fourth uses the minimum value of rainfall that induces major landslides. We used receiver operating characteristic (ROC) analysis to evaluate the predictability of the rainfall thresholds. The fourth method showed the best results compared with the others, and this method provided a good prediction of landslide events with a low error value. The chosen threshold was then applied in the Balai Sabo-ILEWS.

**Keywords:** landslides; early warning system; rainfall threshold; ROC

## 1. Introduction

Landslides are natural phenomena that can cause physical damage and even fatalities, and occur almost all over the world [1,2]. Many countries try to mitigate landslide disasters by using structural methods (e.g., slope stabilization, drainage, vegetation, barriers), nonstructural methods (e.g., early warnings, land-use planning, escape routes, emergency management) or both [2,3]. However, the use of nonstructural treatments is favored due to their ease of application and financial considerations [4]. Among these nonstructural technologies, the most widely applicable measure are landslide early warning systems

(LEWS) [4]. Many countries have developed LEWS with various approaches and coverage areas. Piciullo et al. [5] divided LEWS coverage areas into local and territorial. A system is local when the LEWS is based on only a single or several slopes, and a territorial system covers more extensive areas, such as regional (administrative boundaries of districts and provinces), national (for a country), and global for all of the world [6]. Guzzetti et al. [6] mentioned that from 1977 to 2019, at least 26 locations implemented a LEWS, whether on a national, regional, or global scale.

In Indonesia, landslides are one of the most disastrous natural hazards that frequently occur [7–9]. Rainfall is the primary factor among the various triggering forces creating landslides [10]. Most landslides are caused by rainfall, particularly within the wettest months of the rainy season [11–14]. Rainfall-induced landslides are responsible for approximately 90% of deaths related to slope failure [15]. The rainfall threshold is widely used in the LEWS around the world. Indonesia, therefore, has developed LEWS based on rainfall thresholds on both the regional and local scales [7,16]. The rainfall threshold method provides a quick warning compared with warnings derived from the soil modelling approach. A combination between the empirical rainfall threshold and rainfall measurements is generally the most popular method applied in LEWS [17].

As one of the government agencies that has the task of managing sediment-related disasters, Balai Sabo has developed the Indonesia Landslide Early Warning System (Balai Sabo-ILEWS). Balai Sabo-ILEWS implements the rainfall threshold for a regional-scale LEWS. The warning level given by the Balai Sabo-ILEWS, however, is based on a single rainfall threshold value, which hampers the accuracy of the Balai Sabo-ILEWS. Nevertheless, the information provided by the Balai Sabo-ILEWS is very helpful for increasing public preparedness for landslides [18]. Updating and developing new rainfall thresholds are essential tasks and should be carried out regularly to improve the accuracy of LEWS. More detailed information on the ILEWS can be obtained from https://sda.pu.go.id/balai/tekniksabo/public (accessed on 10 November 2021) and Hidayat et al. [7].

Rainfall data gathered from ground-based observations are highly accurate and are often utilized in landslide research. However, this technique is pricey and has low area coverage due to a lack of ground station density, particularly in mountainous areas. In this study, satellite-derived rainfall products obtained from the Tropical Rainfall Measuring Mission (TRMM) and Global Precipitation Measurement (GPM) have been chosen to overcome this problem [19]. These data can be integrated into nowcasting and forecasting warning systems for landslide risk management if combined with rainfall forecasting (e.g., the rainfall forecasts produced by the European Centre for Medium-Range Weather Forecasts (ECMWF)) [20]. The existing Balai Sabo-ILEWS uses a regional scale with a single rainfall threshold value for all of Indonesia [7]. This system was established in 2017 and is still under further development. The Balai Sabo-ILEWS uses rainfall predictions from the ECMWF as input to predict landslides up to 4 days ahead. In 1 week, this system is run twice, namely on Tuesdays to give warnings for the next 3 consecutive days (Wednesday, Thursday, and Friday) and on Fridays to provide alerts for the next 4 consecutive days (Saturday, Sunday, Monday, and Tuesday) [7].

Differences in the geographical conditions of the Indonesian region mean that the use of a single rainfall value as a threshold leads to inaccurate results. Aldrian and Dwi Susanto [21] stated that rainfall in Indonesia can be divided into three regions, based on their characteristics. Region A starts from south Sumatera and also covers south Kalimantan, Java to Timor Island, Sulawesi, and Irian Jaya. Region B begins at northern Sumatra and continues to northern Kalimantan. Region C consists of the northern part of Sulawesi and Maluku. Region A is the most dominant area because it covers an extensive area [21]. Therefore, this study aimed to improve the accuracy of the Balai Sabo-ILEWS by updating the general threshold that was previously implemented in the system with a more specific threshold that is solely developed for Region A. Statistical approaches based on historical landslide events were used to select the best applicable threshold. Due to the limitations of

the data collected, this study will focus only on Region A, especially Java Island, where most landslides have occurred [22].

## 2. Materials and Methods

### 2.1. Study Area

The location of this research was Java Island. The total area of Java Island is 128.297 km$^2$, consisting of mountainous areas with several active volcanoes and various elevation levels between 0 and 2500 m (Figure 1). Java Island is one of the many islands on the perimeter of the Eurasian Plate in Indonesia, and subduction has had an important impact on its geological history [23]. The physiographic zones identified by van Bemmelen [24] were subdivided into 7 groups based on different ages, processes, and lithology (Figure 2) as follows: (1) From the west to the east of Java Island, the Quaternary Volcanic Zone is defined by a series of volcanic arcs, characterized by alluvial deposits from quaternary volcanoes. (2) The southern mountains zone is characterized by limestone and volcanic rock estimated to be from the Miocene, and there is an endogenous process in the form of uplift in the south of Java Island. (3) The central depression zone, which is the main axis of the island of Java, is formed by the endogenous process. (4) The Kendeng hills have Mio-Pleistocene deposits in part of the middle anticlinal zone extending from west to east. (5) The Randublatung depression, located at the foot of the Kendeng hills, consists of Mio-Plestocene deposits. (6) The Rembang–Madura anticlinorium is composed of limestone rocks formed during the Miocene. (7) The alluvial lowlands formed by the deltaic deposits build the northern coast of Java.

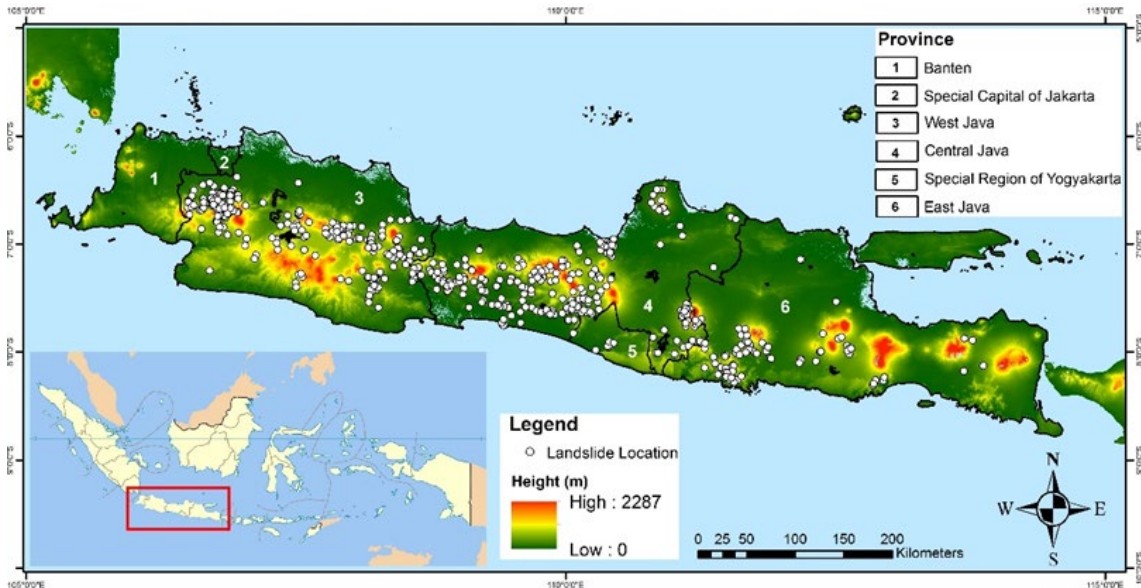

**Figure 1.** Digital elevation model of Java Island with the spatial distribution of landslide events from 2017 to 2020.

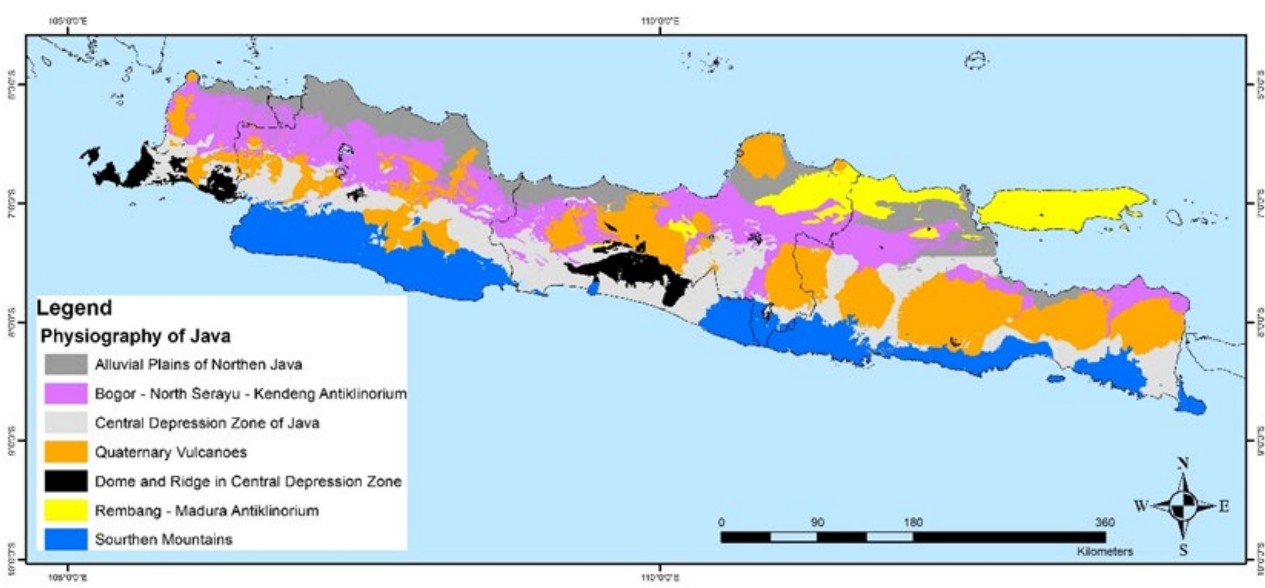

**Figure 2.** Geological map of Java Island obtained from [24].

Java Island has a tropical climate with annual mean air temperatures ranging from 26.4 to 29.6 °C and an average rainfall intensity of 320 mm/month during the rainy season [25]. These conditions make Java Island vulnerable to many types of weather-induced natural disasters, including landslides [26]. Compared with landslide occurrence in the entire nation, Java Island has the highest frequency of landslide occurrence, namely 62.0% of the total landslides in Indonesia, compared with Sumatra (21.2%), Sulawesi (6.7%), and Kalimantan (5.1%) [22]. Aside from this condition, Java Island has the highest population level in Indonesia, with 151.6 million people or 56.10% of the entire population of Indonesia living in this island. The largest population (48.27 million people) resides in the West Java Province, followed by East Java with 40.67 million people, Central Java with 36.52 million, and DI Yogyakarta with 3.67 million people [27]. Java Island has become the center of economic growth in Indonesia with the best infrastructure and industrial facilities; therefore, developing a regional landslide warning system with sufficient accuracy is of utmost importance in this island compared with other islands.

*2.2. Data*

2.2.1. Landslide Events

The landslide data used to build the rainfall thresholds were collected from 2017 to 2020, with a total of 420 landslide events (Figure 1). These landslide inventories were gathered from the government websites responsible for natural disaster management, such as the national and regional disaster management Agencies (BNPB and BPBD, respectively) and from other sources such as digital newspapers, blogs, and technical reports [28]. The data collected should provide information about the location, time, losses of material and life, and the number of affected residents. These data are usually used to provide emergency aid when a landslide occurs [29]. Landslides in this study refer to shallow landslides that were triggered by rainfall. Shallow landslides occur not because of rising groundwater levels but are due to rainfall infiltration in the soil [30].

The landslide events are distributed into several locations based on the area. In total, the landslide events were distributed across four sites (Figure 1). Central Java is the location where most landslides occurred, with 200 landslides or almost half of the total landslide events (48%). West Java and East Java recorded 153 (36%) and 52 (12%) landslides occurrences, respectively. DI Yogyakarta is the location with the fewest landslide events, with only 15 (4%) landslides documented during 2017–2020. This study also categorized several landslide events that affected areas of more than 100 m$^2$ as major landslide events.

Among the 420 landslide events, 23 of them were major landslides. Figure 3 shows examples of landslide events triggered by heavy rainfall in Purworejo and Banyumas Districts, Java.

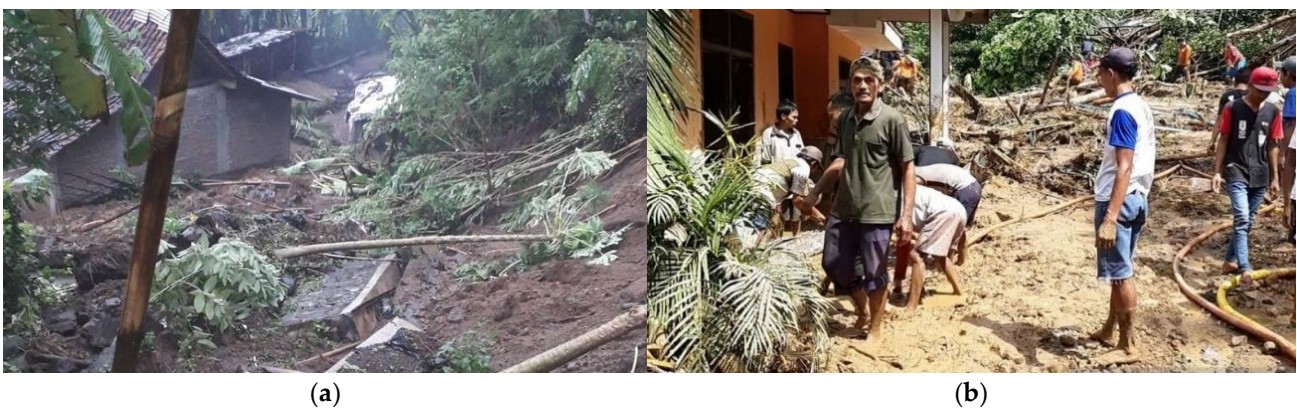

(**a**)　　　　　　　　　　　　　　　　　　　　　　　　(**b**)

**Figure 3.** Landslide events triggered by rainfall in Java Island (**a**) at Purworejo District on 28 October 2020 and (**b**) at Banyumas District on 17 November 2020.

### 2.2.2. Observed Rainfall Data

There is a significant number of global satellite rainfall products, for instance, Global Satellite Mapping of Precipitation (GSMap) Reanalysis, the Climate Prediction Center Morphing Algorithm (CMORPH), PERSIANN-CCS, IMERG, and TRMM [31]. In ILEWS, TRMM data were used as input to develop a flood and landslide monitoring system [32–34]. However, in December 2019, the TRMM mission ended, but it was continued by the GPM mission as its successor. The GPM mission is also a joint project between NASA and JAXA that launched in February 2014 from Tanegashima Space Center, Japan. The GPM has broader global coverage data than TRMM, between latitudes of approximately 65° north and 65° south. The GPM mission carries two instruments, a radiometer called the GPM Microwave Imager and a Dual-Frequency Precipitation Radar [35]. These instruments aim to probe global precipitation characteristics (rain, snow, ice) in a more accurate way and assist in forecasting the extreme events that lead to natural hazards, such as floods, droughts, and landslides [36,37].

For this study, the rainfall dataset used to determine the threshold was derived from the GPM mission. The GPM mission has provided data from 2000 until now. The rainfall data was downloaded from https://giovanni.gsfc.nasa.gov/ (accessed on 8 September 2021) with a spatial resolution of 0.1°. Two types of rainfall data were used in the analysis, namely 1-day and 3-day effective antecedent rainfall. The 1-day and 3-day effective antecedent rainfalls were estimated on the basis of the date when the landslide occurred (from 00.00 to 24.00). For instance, a landslide occurred in West Bandung on 2 January 2018, so the rainfall data from 31 December 2017 to 2 January 2018 were downloaded. The daily rainfall used the rainfall data from the same date of the landslide, in this case, the rainfall data on 2 January 2018. The effective antecedent 3-day rainfall was calculated following Glade et al.'s [38] equation using cumulative rainfall data on the same date of the landslide and the two preceding days' effective antecedent rainfall data, in this case, 31 December 2017 and 1 January 2018.

Other rainfall data required in this study to determine the rainfall threshold were the rainfall events that did not trigger landslides. Following the method proposed by Muntohar et al. [8], the landslide events considered in this study were only the first events, if there were several subsequent landslide events at the same location. To determine a meaningful comparison of the triggering and nontriggering rainfall events, we selected high-intensity rainfall that did not trigger a landslide in the same location as a landslide in the month(s) before the landslide event [39]. For example, if a landslide occurred on 24 April 2017, the 1-day and 3-day rainfall events considered to be nontriggering were high-intensity rainfall in the same location in the month(s) before 24 April 2017. In total,

the number of triggering rainfall events and nontriggering rainfall events collected in this study was 420 and 750, respectively. Figure 4 shows the rainfall data used in this study, with the y-axis being 1-day cumulative rainfall and the x-axis being the 2-day effective antecedent rainfall. The 3-day effective antecedent rainfall is the total of the 1-day and 2-day effective antecedent rainfall.

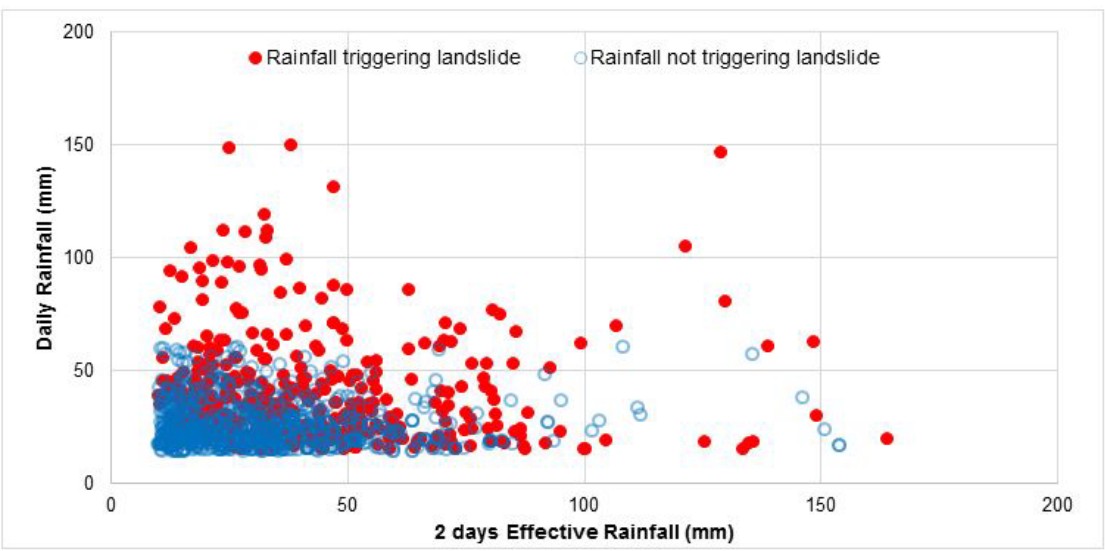

**Figure 4.** Relationship between 2-day effective antecedent rainfall and daily rainfall that triggered and did not trigger landslides from 2017 to 2020.

*2.3. Methods*

2.3.1. Effective Antecedent Daily Rainfall

The rainfall is not only the factor that contributes to slope failure. Pre-existing soil moisture conditions in the area also contribute to the landslide occurrences. As soil moisture measurements are either rarely or not readily available prior to a landslide event, effective antecedent daily rainfall can be considered as an index of soil moisture conditions preceding the event [38,40]. Following Chikalamo et al. [39], who studied rainfall thresholds for Bogowonto Catchment, Central Java, we adopted the effective antecedent daily rainfall model proposed by Glade et al. [38] and Zezere et al. [41]. The effective antecedent rainfall (*AR*) index is calculated as follows:

$$AR_x = kR_1 + k^2R_2 + \ldots + k^nR_n \tag{1}$$

where $AR_x$ is the effective antecedent daily rainfall for day $x$, $k$ is the coefficient of the decay rate of the flood hydrograph curve, $R_1$ is the daily rainfall for the day before day x, and $R_n$ is the daily rainfall $n$ days before day $x$. Unlike Chikalamo et al. [39], who used a range of antecedent period lengths to assess the optimum number of days for calculating the antecedent rainfall, in this study, we only considered 3-day and daily rainfall accumulation. This selection was based on the consideration that the current LEWS uses 1-day and 3-day cumulative rainfall to derive rainfall thresholds [7]. The 3-day effective antecedent rainfall is the sum of daily rainfall and 2-day effective rainfall, following Equation (1), that was calculated before the date of the landslide. Following Chikalamo et al. [39], we also assumed *k* = 0.9, which came from a study of flood hydrograph recessions in the Central Java region. In fact, the decay rate depends on catchment shape and size, vegetation cover, relief, slope gradients, soil type, and the existence of natural or artificial lakes [38].

### 2.3.2. Determination of Rainfall Thresholds

The rainfall corresponding to each landslide event in the inventory was obtained from the GPM data by extracting the daily rainfall value from the GPM grid cell covering the event's location on the event date. The same procedure was applied to obtain rainfall estimates for up to 3 days preceding a landslide event to calculate the antecedent rainfall. We derived the new rainfall thresholds by analyzing the relationship between the 1-day rainfall during the landslide event and the 3-day effective antecedent rainfall. This relationship was shown by plotting the daily cumulative rainfall on the day of the landslide events in each region against the 3-day effective antecedent rainfall. Afterwards, without using rigorous probabilistic analysis, a regression line was drawn from the scatter plot to obtain the rainfall thresholds (Figure 5a,b) using four methods. The first method is the existing rainfall threshold implemented in the Balai Sabo-ILEWS. The second uses the same approach as the existing Balai Sabo-ILEWS but with new data collected from 2017 to 2020. A threshold line was drawn based on the average value of the 1-day and 3-days effective antecedent rainfall [7]. The third method connects the lowest values of the rainfall events that triggered landslides [8]. Lastly, the fourth method refers to Hong et al. [42], in which the threshold line was drawn based on the minimum value of major landslides.

### 2.3.3. Performance Analysis of Rainfall Thresholds

All the rainfall thresholds derived by all methods were evaluated using the receiver operating characteristic (ROC). The ROC is a two-dimensional plot describing the performance of the classifier system [43], using the contingency table and the area under the curve (AUC). The contingency table consists of four conditions based on the model predictions, and the occurrences or non-occurrences of a specific event are categorized as true positive (TP), true negative (TN), false positive (FP), and false negative (FN). Positive and negative refer to the model predictions, while true and false refers to the validity of the model predictions and real events [44]. In application, TP occurs when the rainfall is above the threshold and a landslide occurs. TN occurs when the rainfall is below the threshold and a landslide does not occur. FP occurs when the rainfall is above the threshold but a landslide does not occur. FN occurs when the rainfall is below the threshold but a landslide occurs (Table 1). In summary, there are two conditions: correct predictions (TP and TN) and wrong predictions (FP and FN) [8]. The performance of the thresholds was measured based on several statistical indices obtained from calculation, as shown in Table 1. The statistical indices are shown in Table 2.

The area under the curve (AUC) was used to evaluate the ability of the classifier system to discriminate the actual condition of landslide events in the field [43]. To create the AUC, we have to draw the ROC curve by plotting the sensitivity on the Y-axis and the 1-specificity on the X-axis for all possible thresholds. The capability levels of the classifier system based on the AUC value are presented in Table 3 [43].

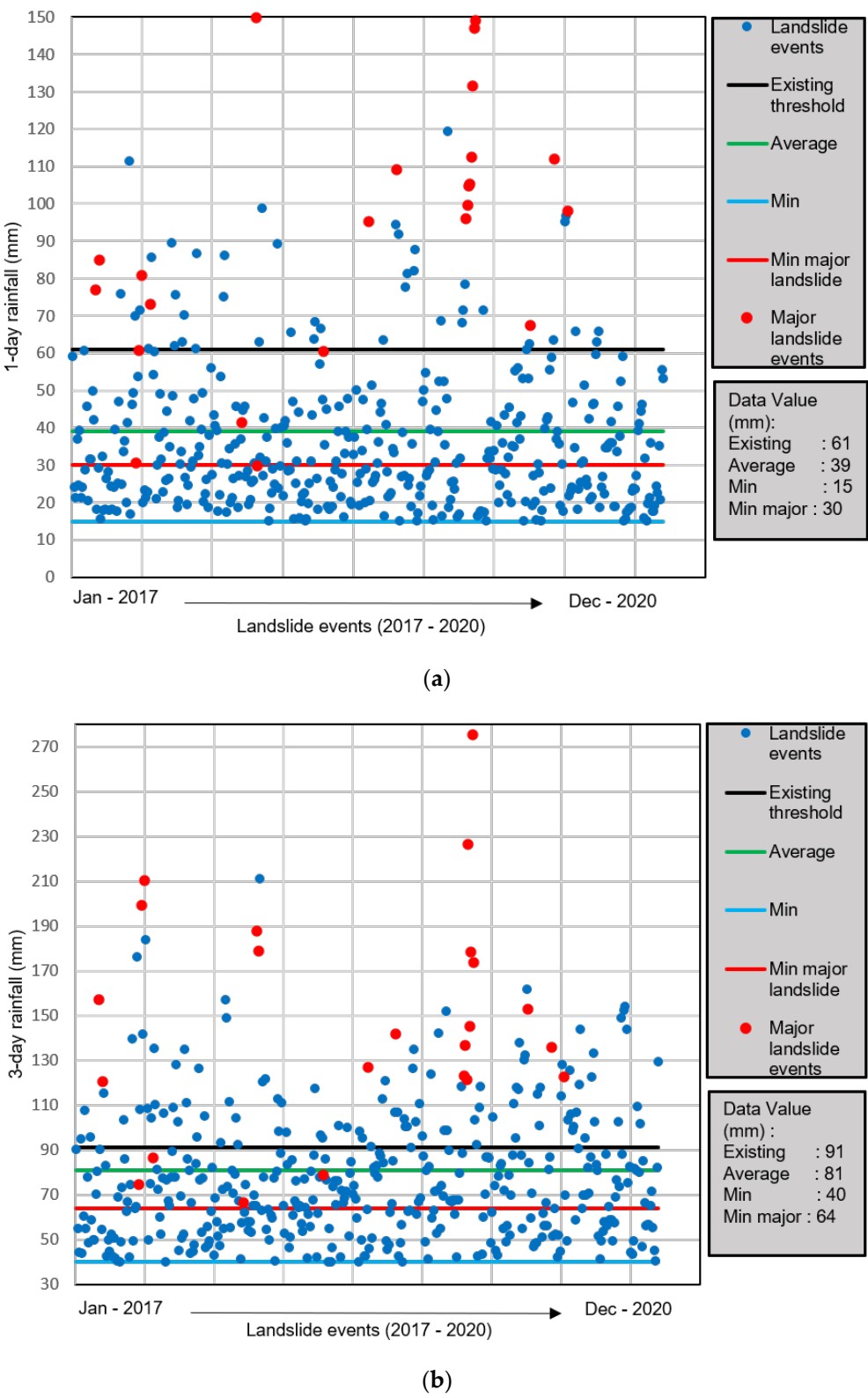

**Figure 5.** (**a**) One-day cumulative rainfall threshold. (**b**) Three-day effective antecedent rainfall threshold.



**Table 1.** Contingency table.

| Model Predictions | Landslide Events | |
|---|---|---|
| | **Yes** | **No** |
| ≤Threshold | TP | FP |
| >Threshold | FN | TN |

**Table 2.** Statistical indices used to measure the performance of the thresholds.

| Statistical Indices | Description | Equation |
|---|---|---|
| Sensitivity/true positive rate (TPR) | The proportion of positive cases of landslide events that are correctly detected by the threshold | $TPR = \frac{TP}{TP+FN}$ |
| Specificity/true negative rate (TNR) | The proportion of negative cases of landslide events that are correctly detected by the threshold | $TNR = \frac{TN}{TN+FP}$ |
| Accuracy | The proportion of correct predictions overall | $Acc = \frac{TP+TN}{TP+TN+FP+FN}$ |

**Table 3.** AUC value classifications.

| AUC Value | Descriptions |
|---|---|
| 0.5 | No discrimination, random guesses |
| 0.5 < AUC ≤ 0.6 | Poor discrimination |
| 0.6 < AUC ≤ 0.7 | Acceptable discrimination |
| 0.7 < AUC ≤ 0.8 | Excellent discrimination |
| 0.9 < AUC | Outstanding discrimination |

## 3. Results

### 3.1. Rainfall Threshold Results

Following the methods to estimate the threshold lines as explained in Section 2.3.2, we derived four rainfall thresholds for all 1-day cumulative rainfall (Figure 5a) and 3-day effective antecedent rainfall (Figure 5b). The four rainfall threshold values in the 1-day and 3-day effective antecedent rainfall showed the same pattern. The biggest threshold is the existing threshold, with a value of 61 mm and 91 mm for the 1-day and 3-day effective antecedent rainfall, respectively. This is followed by the average threshold, with results of 39 mm and 81 mm, then the minimum for major landslide threshold with values of 30 mm and 64 mm. The last is the smallest landslide threshold, which had the lowest values of rainfall that triggered landslides, with threshold values of 15 mm and 40 mm for 1-day and 3-day effective antecedent rainfall, respectively.

### 3.2. Performance Analysis

#### 3.2.1. ROC Curve

The performance analysis results from using the ROC curve for both 1-day and 3-day effective antecedent rainfall are shown in Figure 6. This indicates that the 3-day effective antecedent rainfall produces higher performance (AUC = 0.73) than the 1-day cumulative rainfall (AUC = 0.70). The difference between the 1- and 3-day cumulative rainfall, however, is relatively small. AUC values above 0.7 imply that the model shows excellent discrimination (Table 3), meaning that the thresholds derived by the classifier system have a very good level of accuracy and are not just random guesses.

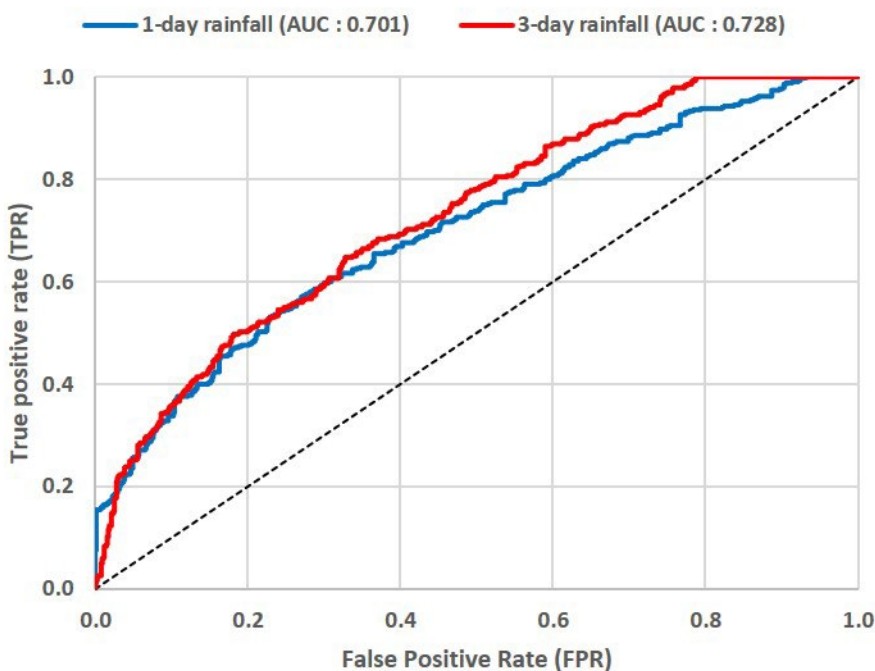

**Figure 6.** AUC results for 1-day and 3-day effective antecedent rainfall.

### 3.2.2. Confusion Matrix and Statistical Indices

The results of the confusion matrix and statistical indices for each threshold in this study are summarized in Table 4. In total, we used 420 landslide events and 1170 rainfall events for the 1-day and 3-day effective antecedent rainfall. The statistical results are shown as the average 1-day and 3-day effective antecedent rainfall. Statistical values above 0.5 are considered to be able to provide an explanation of the performance, and a value above 0.5 is categorized as "good" and one below 0.5 is categorized as "not good" [31].

TPR and TNR

TPR and TNR describe the ability of the threshold to predict actual landslide events and actual landslide nonevents, respectively. Thus, higher TPR and TNR results mean that the threshold performs better. Based on Table 4, the TPR values, ranked from the highest to the lowest, were obtained by the thresholds calculated using the third, fourth, second, and first methods. The third and fourth methods showed "good" TPR values with values higher than 0.5, 0.99 and 0.60, respectively. In contrast, the second and first methods showed "not good" TPR values of 0.38 and 0.21, respectively. The TNR values, starting from the highest to the lowest values, were ranked as the first method with a TNR value of 0.97, the second method with a TNR value of 0.89, the fourth method with a TNR value of 0.71, and the third method with a TNR value of 0.14. Thus, all methods yielded "good" performance except for the third method, which showed "not good" performance. If we combine the TPR and TNR values, then the highest TPR+TNR values are obtained by the fourth method with a total value of 1.31, followed by the second method, the first method, and the third method, with total values of 1.27, 1.18, and 1.13, respectively.

Accuracy

Accuracy is the correct percentage prediction of a threshold regarding whether landslides occur or not, compared with all predictions generated by the threshold [28]. The higher the accuracy, the better the threshold. Table 4 shows that other than the third method, all the methods produce good prediction accuracy scores, with a small difference in accuracy. The second method had the highest accuracy of 0.71, followed by the first method (0.7) and the fourth method (0.68), while the third method had the lowest accuracy of 0.45.

**Table 4.** Result of the confusion matrix and statistical indices.

| Method | Threshold Line | Duration | Threshold (mm) | Contingency Table * | | Av. ** TPR | Av. TNR | Av. Accuracy |
|---|---|---|---|---|---|---|---|---|
| 1st | Existing threshold | 1 Day | 61 | 59 | 1 | 0.21 | 0.97 | 0.70 |
| | | | | 361 | 749 | | | |
| | | 3 Days | 91 | 120 | 44 | | | |
| | | | | 300 | 706 | | | |
| 2nd | Average | 1 Day | 39 | 155 | 79 | 0.38 | 0.89 | 0.71 |
| | | | | 265 | 671 | | | |
| | | 3 Days | 81 | 168 | 93 | | | |
| | | | | 252 | 657 | | | |
| 3rd | Minimum | 1 Day | 15 | 417 | 693 | 0.99 | 0.14 | 0.45 |
| | | | | 3 | 57 | | | |
| | | 3 Days | 40 | 417 | 590 | | | |
| | | | | 3 | 160 | | | |
| 4th | Minimum of major landslides | 1 Day | 30 | 229 | 186 | 0.60 | 0.71 | 0.68 |
| | | | | 191 | 564 | | | |
| | | 3 Days | 64 | 272 | 245 | | | |
| | | | | 148 | 505 | | | |

* Includes the values of TP, FP, FN, and TN, referring to Table 1. ** Average.

Overall Statistical Scores

Based on the results from Table 4, we summarized the general performance of the thresholds in Table 5. We categorized scores above 0.5 as "good" and those below 0.5 as "not good" (as explained in Accuracy Section). The best threshold was identified for the threshold generated by the fourth method because it was the only threshold that showed good performance in all categories. The fourth method can provide good predictions of landslide events and landslide nonevents. In addition, this method also has a small error prediction rate compared with the actual landslide events. The first and second methods have similar performance. The thresholds derived from these approaches have poor performance for predicting landslides, even though the thresholds have good results for predicting landslide nonevents. Both methods, however, have good prediction accuracy. The last is the third method, which yields a good result for predicting landslide events but it has poor performance in estimating landslide nonevents. Moreover, the accuracy of this threshold is below 0.5, and it was categorized as a method that produces poor prediction performance.

**Table 5.** The general performance of the thresholds.

| Method | Threshold Line | TPR | TNR | Accuracy |
|---|---|---|---|---|
| 1st | Existing threshold | Not Good | Good | Good |
| 2nd | Average | Not Good | Good | Good |
| 3rd | Minimum | Good | Not Good | Not Good |
| 4th | Minimum of major landslides | Good | Good | Good |

## 4. Discussion

### 4.1. Worldwide Rainfall Thresholds

In Figure 7, we plot other regional rainfall thresholds collected from 2018 to 2021 from all over the world proposed by [8,39,45–47] in a power law as a comparison. All of those thresholds use a similar condition/approach to our study (regional rainfall threshold in

a mountainous area and only considering shallow landslides). We also plot the existing Balai Sabo-ILEWS threshold (first method), that proposed by Hidayat et al. [7], and the proposed threshold derived from the best method, which is the fourth method. This plot only considers the minimum 1-day and 3-day rainfall thresholds following the rainfall threshold used in this study. Figure 7 shows that the proposed threshold in this study lies in the middle of other thresholds. This indicates that the selected threshold is in good agreement with the other thresholds.

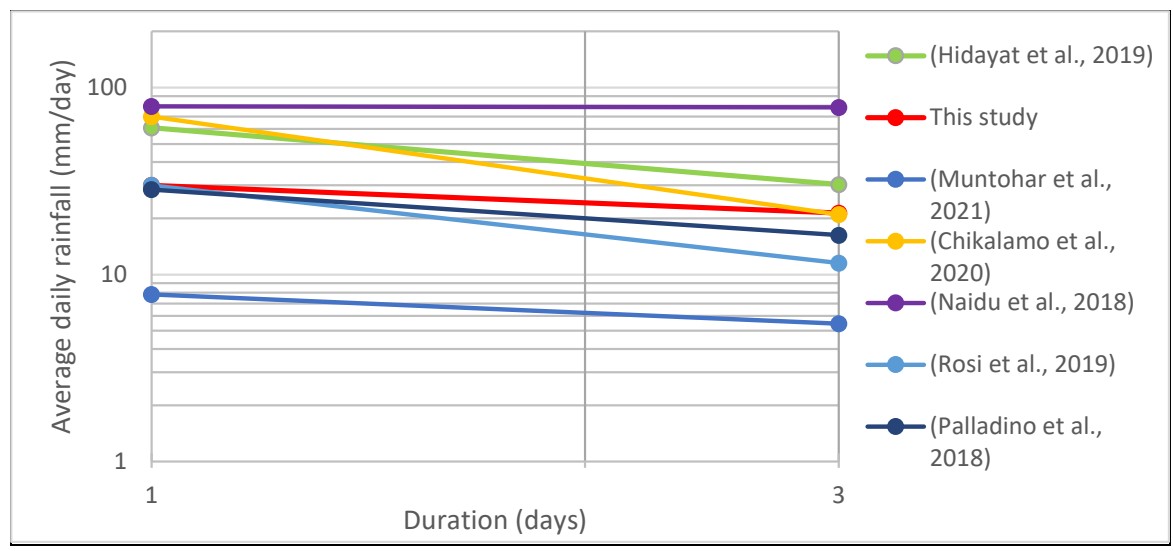

**Figure 7.** Worldwide rainfall thresholds triggering landslides, collected from 2018 to 2021 [8,39,45–47].

The landslide thresholds derived from average daily rainfall vary from less than 10 mm to more than 70 mm. This indicates that the landslide threshold is highly dependent on the location, climate, and method used for determining the threshold line [48]. Highland regions with natural steep slopes and lowland areas with artificial slopes will have a different rainfall intensity prior to a landslide, resulting in a different rainfall threshold. Furthermore, developing a landslide threshold in a certain location must consider the differences in seasons, climate, land cover, and soil conditions compared with other locations, which will lead to a different threshold value, even though the location being reviewed is the same.

In this study, we only reviewed the general threshold for the entire region of Java Island without considering differences in the local conditions, such as differences in seasons, land cover, and soil conditions. The threshold in this study was used for all of Java Island during either the rainy season or dry season. Compared with the existing threshold (the first method), the newly proposed threshold (the fourth method) showed better prediction performance but a lower precipitation threshold. The 1-day cumulative rainfall was reduced by more than half, from 61 mm/day to 30 mm/day, and the 3-day cumulative rainfall was reduced by almost 30%, from 91 mm/3 days to 64 mm/3 days. This condition occurred because the trend of rainfall events that trigger landslides in the research location is decreasing for the period of 2017–2020, which means that lower rainfall intensity is needed to trigger a landslide event. However, we should note that the average of the rainfall events that triggered landslides in 2017–2020 in terms of the 1-day and 3-day cumulative rainfall is 39 mm and 81 mm, respectively, and both of these still lie below the existing thresholds. Our result suggests that the use of existing thresholds should be revisited, and the use of newly developed thresholds must be urgently implemented in the Balai Sabo-ILEWS.

### 4.2. Implementation in the Balai Sabo-ILEWS

Rainfall threshold data combined with susceptibility maps are often used to generate LEWS and are proven to give good results [49–52]. A susceptibility map is used to predict the landslides' locations, while the rainfall threshold is used to estimate the time of landslide events [49,53]. This method was also applied in this study by combining the selected rainfall threshold with Indonesia's susceptibility map. Indonesia's susceptibility map was established by the Geological Agency of Indonesia in 2017 using a heuristic and statistical approach [54]. With a resolution of 1:250.000, the susceptibility map divides Indonesia into four zones namely: very high (red), moderately high (orange), low (blue) and very low (green). Several parameters were used to build this map, such as the slope, soil type, geological structure, land use, and landslide history [54]

The existing Balai Sabo-ILEWS provides information about landslide warnings only for the locations that are indicated on the landslide susceptibility maps as "moderately high" and "very high" [1]. Other conditions such as "very low" and "low" are assumed to be always stable. The recent system assumes that the "moderately high" and "very high" regions have the same landslide probability, even though it is not the case. The "very high" regions have a higher landslide probability than the "moderately high" regions. In this study, we propose a new approach to be implemented in the Balai Sabo-ILEWS that considers the differences in rainfall classification and the landslide susceptibility map. The landslide susceptibility map is shown in Figure 8.

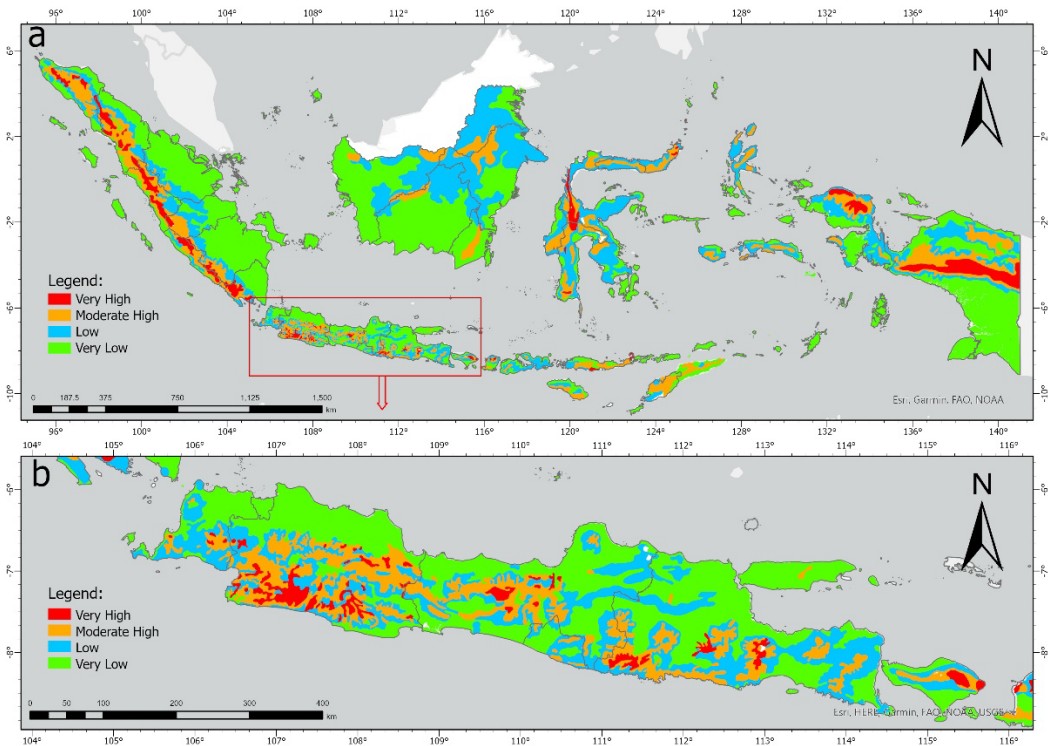

**Figure 8.** Landslide susceptibility map implemented in the Balai Sabo ILEWS. (**a**) Landslide susceptibility map at the national scale of Indonesia, and (**b**) detailed susceptibility map for Java Island.

To define landslide warning levels, we classified the warning levels based on the amount of 1-day and 3-day effective antecedent rainfall and the landslide susceptibility map. The rainfall classification is divided into three conditions. A very high rainfall classification is applied if 1-day and 3-day effective antecedent rainfall of more than 30 mm and 64 mm occurs, respectively. A high rainfall classification is applied if it meets only one of these conditions, and low rainfall classification is given if it does not meet both conditions. The warning level gives four levels of warning in different classes. Warning classes are

intended to provide better information for the stakeholders. Following Park et al. [55] and Pradhan et al. [50], warning levels, from the lowest to the highest, are indicated by green (Null), yellow (Watch), orange (Attention), and red (Alert) (Figure 9). "Null" is used if there is no warning, "Watch" as a preliminary warning, "Attention" as a precautionary warning, and "Alarm" as a warning that a landslide may occur at any time.

**Figure 9.** Definition of the warning system.

Rainfall with a low classification will trigger a "Watch" warning in the regions located in the very high landslide susceptibility areas. The rest (moderately high, low, and very low), receive no warning or "Null". Rainfall classified as high will trigger "Attention" and "Watch" warnings for regions located in the very high and moderately high risk areas, respectively. For low and very low risk regions, there are no warnings. The last, (very high) rainfall classification will trigger "Alarm", "Attention", and "Watch" warnings for regions located in the very high, moderately high, and low landslide-susceptibility areas, respectively. The very low regions receive no warning, whatever the rainfall classification is. Through this method, the same precipitation classification in different landslide-prone locations will produce different warning levels.

We applied this new approach in Banjarnegara district to find out the performance. This location is within Central Java, where almost half of the landslide events in this study occurred. The total number of landslides in Banjarnegara during 2017–2020 was 15 events. The results show that 23 "Alarm" warnings were recorded, 122 for "Attention", 373 for "Watch" and 578 for "Null". All the actual landslide events could be predicted by the system, although the accuracy was 65%. Among the 15 landslide events, most of them occurred when there was an "Alarm" warning and only two landslide events occurred at "Attention" and one landslide event at "watch". Another issue is the number of "Attention" and "Watch" warnings, which were quite numerous in our opinion. This condition occurred because most of Banjarnegara area was categorized as "very high" in the susceptibility map, which means that low-intensity rainfall was enough to trigger the warning. In general, the results in the test area were quite good, even though there was an over-prediction of warnings. The use of a susceptibility map with a more detailed scale is also expected to improve the results of this system further.

## 5. Conclusions

This study derived a rainfall threshold that will be implemented in Java Island using four methods. The first method used the existing threshold in the Balai Sabo-ILEWS. The second method applied the average value of rainfall as the threshold. The third and fourth methods used the minimum rainfall of all the landslide events and the minimum rainfall for only the major landslide events, respectively. The fourth method had the best performance because it showed "good" results for all statistical indices (TPR, TNR, and accuracy). The

first and second methods showed "not good" scores for TPR, while the third method produced "not good" scores in the TNR and accuracy categories. The result shows that the new rainfall threshold proposed in this study (the fourth method) has good performance statistically. The threshold gives good predictions for landslide occurrence, with a low error level. In general, the new rainfall threshold is capable of being used as a reference for the Balai Sabo-ILEWS.

This study is a part of efforts to improve the existing Balai Sabo-ILEWS's performance. The newly developed rainfall threshold for the Java area has recently been implemented in the system, and we will gradually apply different rainfall thresholds to all other regions in Indonesia. Moreover, we will also issue warning levels based on different rainfall and landslide susceptibility classifications. Further data collection, especially for landslide occurrences in regions other than Java Island, is still needed to improve the accuracy of the Balai Sabo-ILEWS in the future.

**Author Contributions:** R.A.Y. worked on conceptualization, writing, methodology, data analysis, and visualization; A.R., F.F., and S.J.S. worked on conceptualization, validation, review and editing, supervision, and improving the paper; A.S., R.S., and A.N.H. worked on writing, data analysis, and visualization; E.P. worked on writing, data analysis, and data curation; R.H., A.M., A.S.M., V.G.J., C.J.v.W., and B.V.d.B. worked on discussion, investigation, and supervision; B.W.R. worked on software and visualization. All authors have read and agreed to the published version of the manuscript.

**Funding:** This research was supported by UGM.

**Institutional Review Board Statement:** Not applicable.

**Informed Consent Statement:** Not applicable.

**Data Availability Statement:** Data sharing is not applicable.

**Acknowledgments:** The authors would like to thank all the staff of the Balai Teknik Sabo and the Department of Civil and Environmental Engineering, UGM, who have helped to carry out this research. Thanks also to the lecturers in ITC, Twente University and the Department of Civil Engineering, UMY for discussing a problem in this research.

**Conflicts of Interest:** The authors declare no conflict of interest.

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
