# Peer review of "Revised Rainfall Threshold in the Indonesian Landslide Early Warning System"

_geosciences, doi:10.3390/geosciences12030129_

Round 1
Reviewer 1 Report
The present study used four methods to determine the rainfall threshold that triggers landslides and compared their prediction results. The topic is interesting and useful for the mountainous zone management. I have several suggestions provided below for the authors before the paper can be accepted.
- In Line 176, the symbol for the dependent and independent variables should be provided as the variables firstly appear. For example, the antecedent daily rainfall AR…. Revise all similar uses.
- The italic font and non-italic font are different. Please make sure if your notation is italic. For instance, the notations in Lines 178 and 179 are non-italic, but notations in Eq.(1) are italic. Check all similar mistakes.
- Line 240, where is the verb?
- In Eq.(2), the “X” should use “Ë‘”.
- Line 263, “It’s” can not be used in an academic paper. Check all similar mistakes.
- Lines 324 to 326, do not use similar terms (a good performance) in the same sentence. Please use alternative terms for every sentence in the paper.
- As you mentioned in the paper, the worldwide rainfall threshold may vary for different factors. So, what is your purpose to compare these thresholds?
- In Figure 9, how do the government obtain “Landslide susceptibility map”? Please briefly describe it.
- In Figure 5, how do you determine the major landslide events? Please explain it. 
Reviewer 2 Report
The paper deals with an interesting topic, early warning for rainfall-induced landslides, in relation to a case study in Indonesia. In particular, the authors propose the use of new rainfall thresholds in a test area in lieu of the currently adopted one, which is valid nationwide. Although the innovation in the paper is rather limited, as the authors do not propose any novel procedure to analyze the rainfall and landslide data, the manuscript may be considered, after revisions, for publication (it may be considered a technical note rather than a research article) because it presents previously unpublished data to upgrade an existing system in a test area.
The manuscript needs major revisions, also needing further processing of data. The major issues to address are the following.
[Section 2.2.2]. Rainfall triggering and not triggering landslides is (correctly) considered in the analysis. However, only a subset of rainfall not triggering landslides is included. What is the logic of this choice? All the non-triggering rainfall events should have been included. Revise the analyses accordingly. Moreover, the authors do not clearly explain how they selected the rainfall events to include. The sentence they write (“rainfall events at the same landslide location that occurred month(s)/year(s) before the landslide event”) is ambiguous and does not allow reproducibility of the procedure. Clearly explain how to identify and select a rainfall event that did not produce landslides. At what times, the 1-day and 3-day cumulated rainfalls are computed? Considering the spatial representativeness of the rainfall dataset, what is the implication of neglecting the (extremely numerous) locations, i.e., GPM cells, where landslides did not occur?
[Sections 2.2.2 and 2.3.1]. What is the 3-day antecedent rainfall used in the analysis? It is not clearly defined. Is it the total sum of 3 days of rainfall, or a weighted sum (antecedent rainfall index) computed using Eq. 1? If the latter is true, the Authors should say it consistently throughout the paper, not using the term 3-day cumulative rainfall. Alternatively, Eq. 1 and accompanying text should be deleted, to avoid confusion. Moreover, Eq. 1 is also wrong because if the pedix 0 is used to identify the day of the event, the sum should go from 0 to 2 (not from 1 to 3).
[Section 2.3.3] Five statistical indicators, derived from results expressed in a contingency table, are introduced (and later used) to evaluate the performance of the thresholds. Two of them are a useless repetition, indeed TPR = 1 – FNR and TNR = 1- FPR. The authors should decide whether to use TPR or FNR (not both) and TNR or FPR (not both), otherwise when the results are reported (e.g., Table 5) they only generate confusion.
[Section 3.1] Results are reported also as I-D curves, limited to the D space going from 1 day to 3 days. I consider it inappropriate because it is either confusing (you cannot derive and I-D curve from 2 points) or plainly “wrong” if Eq. 1 is indeed used (see my second comment).
[Table 5] Delete the columns of the redundant statistical indicators (see my third comment). The last two columns refer to “sums” that have not a statistical meaning. Avoid using them.
[Section 4.2, lines 380-385] The three rainfall conditions refer to the following LOGIC OPERATORS applied to the rainfall thresholds of the two variables (1-day and 3-day rainfall): AND for “very high”, OR for “high”, and NONE for “low”. This implementation confirms that it is useless and confusing to refer to I-D curves (wherein only the AND and NONE logic operators can be used) in the previous sections of the paper.
[Section 4.2] What happens if the system proposed (Figure 10) is applied in the test area for the time of the -analysis (2017-2020)? How many times would the “alarm”, “attention” and “watch” warning levels would be attained? If the implementation of the thresholds in the LEWS system is introduced, its results must also be presented.
Round 2
Reviewer 1 Report
I am satisfied with the revision. I do not have further comment.
Reviewer 2 Report
The Authors adequately addressed all the points raised by the reviewers. The revised paper can be accepted.